# Delayed Adverse Events after Procedural Sedation in Pediatric Patients with Hematologic Malignancies

**DOI:** 10.3390/medicina58091208

**Published:** 2022-09-02

**Authors:** Jin Joo, Sanghyuck Yu, Hyun Jung Koh

**Affiliations:** Department of Anesthesiology and Pain Medicine, College of Medicine, The Catholic University of Korea, Seoul 06591, Korea

**Keywords:** adverse event, hematological malignancies, pediatric, procedural sedation, propofol, ketamine

## Abstract

*Background and objectives:* Procedural sedation for bone marrow examination (BME) and intrathecal chemotherapy (ITC) is necessary for pediatric patients with hematological malignancies. There has been no report on adverse events after discharge from the recovery room. This retrospective study evaluated the types and incidences of delayed adverse events among pediatric patients scheduled for BME or ITC under deep sedation in a single center for 3 years. *Materials and Methods:* The patients were divided into two groups: inpatients (group I) and outpatients (group O). All patients were managed during the procedures and the recovery period. In total, 10 adverse events were assessed; these occurred 2 h (T1, acute), 12 h (T2, early), and 24 h (T3, delayed) after the procedure. The duration of each adverse event was also recorded and was classified as 2 h (D1), 12 h (D2), or 24 h (D3). The data of 263 patients (147 inpatients and 116 outpatients) who met the inclusion criteria were analyzed. *Results:* The overall incidence of adverse events was statistically significant difference: 48.3% in group I and 33.6% in group O (*p* = 0.011). The rates of adverse events at T1 and T2 were significantly different between groups I and O (42.8% vs. 11.2% and 7.5% vs. 20.7%, respectively) (*p* < 0.001). The adverse events were mostly of D1 or D2 duration in both groups. Patients with a higher proportion of ketamine in a propofol–ketamine mixture had a significantly higher proportion of adverse events at T1 (34.6%), as compared with those with a mixture with a lower proportion of ketamine (21.1%) or propofol alone (17.9%) (*p* = 0.012). *Conclusions:* The most common adverse events were dizziness or headache; typically, they did not last longer than 12 h. The propofol-ketamine combination with a higher proportion of ketamine seems to produce more adverse events within 2 h after the procedure. Nevertheless, all sedative types appear safe to use without additional management.

## 1. Introduction

Procedural sedation in pediatric patients has been the standard of care for relieving anxiety and pain and for producing immobility to optimize conditions for procedures for the last two decades [1,2]. Pediatric patients with hematological malignancies in particular undergo bone marrow aspiration with biopsy (bone marrow examination, BME) and spinal tap with intrathecal chemotherapy (ITC) repeatedly, which can cause psychological trauma. Thus, procedural sedation for BME and ITC is necessary for pediatric patients with hematological malignancies. 

Levels of sedation may vary depending on the needs of the children and the type of procedure. Both BME and ITC are invasive but performed within a short period of time. Therefore, agents producing a deep level of sedation and promoting faster recovery are preferred [3]. Various sedatives are used in pediatric sedation [2,4]. Among these, ketamine and propofol, which induce more profound sedation, are also frequently used together with analgesics to reduce the pain accompanying BME and ITC. 

Several studies have reported the safety and efficacy of procedural sedation [5,6,7,8]. In most cases, the focus is only on the major side effects, i.e., respiratory and hemodynamic adverse effects, such as apnea and bradycardia, during the procedures, and on immediate (within 2 h) post-procedural adverse effects such as headache, dizziness, nausea, and vomiting [9,10,11,12,13]. However, the safety of sedation is not guaranteed, and adverse events may occur 2 h after procedures performed on pediatric patients in an outpatient clinic. In other words, various minor adverse events that may occur late after sedation are overlooked. There are no reports on adverse events after sedation using propofol and/or ketamine in patients who had undergone BME and ITC following discharge from the recovery room. Thus, in this retrospective study, we evaluated the types and incidences of delayed adverse events after sedation using propofol and/or ketamine in these patient groups.

## 2. Methods

### 2.1. Study Population and Ethical Approval 

We reviewed the medical records of pediatric patients who underwent BME and ITC under deep sedation by an anesthesiologist due to hematological malignancy at Seoul St. Mary’s Hospital from March 2019 to February 2022. Hematological malignancy included were classified into two categories; acute lymphoblastic leukemia (ALL), acute myeloblastic leukemia (AML). Data were extracted from the hospital’s electronic chart database. Patients younger than 2 years old and who required airway management other than manual mask ventilation, with complications including organ failure due to their disease were excluded. Among inpatients, in case of the length of hospital stay was extended for treatment other than BME or ICT, those were also excluded. Data on sex, age, height, body weight, duration of sedation, duration of the procedure, type of sedative agent administered, type of procedure performed, amount of fentanyl administered, and type, onset time and duration of adverse events were collected. Patients aged <15 years were defined as children. 

The Institutional Review Board of Seoul St. Mary’s Hospital, and Ethical Committee of Catholic University of Korea approved this study (Ethical Committee N° KC22RIS10323), and waived the requirement for informed consent because of the retrospective nature of the study

### 2.2. Procedure and Intervention

#### 2.2.1. Anesthetic Methods

All pediatric patients were scheduled to undergo BME and ITC under deep sedation at our hospital and were managed as follows. The patients were allowed to eat solid food and drink clear liquid until 8 and 2 h before the procedure, respectively. Intravenous access was established with a 20–24-gauge angiocatheter. Electrocardiography and pulse oximetry were continuously monitored, and noninvasive blood pressure was measured when the patient arrived at the sedation room, and at the end of the procedure. After monitoring, ramosetron 6 μg/kg and 1 µg fentanyl (for ITC) or 1.5 µg/kg fentanyl (for BME and combination procedures) were administered. Then, the following agents were administered for sedation: 7.5 mg propofol and 2.5 mg ketamine in 1 mL (ratio of 3:1; PK25), 8 mg propofol and 2 mg ketamine in 1 mL (ratio of 4:1; PK20), or propofol alone. Initially, 0.25 mL/kg PK25/PK20 or 1.5 mg/kg propofol was administered. In order to avoid the use of the same drug and to find an effective drug for each patient, the sedatives were applied based on the preference of anesthesiologists. The three sedatives PK25, PK20, and P was selected by three experts’ choice and used by them each. All sedation was undergone according to the sedative protocol with reference to the previously used drug combination [14,15,16]. Then, 1–2 mL of each sedative agents was additionally administered, and in some cases, 5–10 µg fentanyl was added if patient movements interfered with the procedure after the initial bolus administration. Manual mask ventilation with Jackson Ree breathing circuit was used to ensure patient safety. 

#### 2.2.2. Postoperative Management

After the procedure was completed, each patient went through recovery in the sedation recovery room located right next to the sedation room (Group I) after the anesthesiologist confirmed that the patient had no respiratory depression. Then, inpatients were transferred to their room in the ward and outpatients (Group O) were discharged and went to their homes after they were fully awake and allowed to lie in bed for 2 h to promote hemostasis in BME cases and for 4 h to prevent headache in ITC cases. The duration in recovery room, up to 4 h is solely due to the nature of of the procedure and it is not associated with the occurrence of respiratory or hemodynamic complication. 

### 2.3. Data Collection and Evaluation

In total, 10 adverse events (Table 1) were recorded 2 h (T1, acute) based on the stay in recovery room and direct side effect of sedatives, 12 h (T2, early) in consideration of half-life of sedatives and residual drug elimination time, and 24 h (T3, delayed) after considering the duration of the existing abnormal reaction in T1 and T2. In all records, each procedure was tracked retrospectively on a checklist of electrical records. For this study, Table 1 checklist was reorganized with additional timing and duration. Adverse events were recorded by recovery nurse during recovery room until 2 h (T1). However, after discharge from the recovery room, they were recorded by the ward nurse in charge of the patient in Group I, while the researcher recorded the adverse events described by the caregiver, i.e., their parents, on the phone until 24 h after discharge at two times, 12 h (T2) and 24 h (T3). The duration of adverse events was also recorded and classified as 2 h (D1), 12 h (D2), or 24 h (D3). We analyzed the main adverse reactions in each group in terms of frequency, timing, and duration. In addition, the effects of the various combinations of sedatives were compared.

### 2.4. Statistical Analysis

All statistical analyses were performed using SPSS software (ver. 20.0; IBM Corp., Armonk, NY, USA). To analyze demographic data, the Chi-square test or Fisher’s exact test for categorical variables and the t-test for continuous variables were used. The incidence rates of the adverse events were compared between the groups using the Chi-square test. Data are presented as means with standard deviations for continuous variables or numbers and percentages for categorical variables. Adverse events occurrence according to additional sedative administration was compared using the Chi-square test or Fisher’s exact for categorical variables. A *p*-value < 0.05 was considered statistically significant.

## 3. Results

### 3.1. Demographic Data (Baseline Characteristics) 

The data on 263 patients (147 inpatients and 116 outpatients) who met the inclusion criteria were analyzed. The patients in group I underwent 74 BME, 24 ITC, and 48 combined procedures, while those in group O underwent 56 BME, 11 ITC, and 49 combined procedures. The patients in group I received 107 PK25, 25 PK20 and 15 propofol treatments, while patients in group O received 46 PK25, 46 PK, and 24 propofol treatments (Table 2). 

### 3.2. Adverse Events in Each Group

The overall incidence of adverse events was significantly different between the groups (48.3% in group I and 33.6% in group O; *p* = 0.011).

The incidence rates of adverse events at T1 and T2 were significantly different between groups I and O (42.78% vs. 11.21% and 7.48% vs. 20.67%, respectively). There was no difference at T3 (Figure 1). In most cases, no intervention was necessary against the occurrence of adverse events in both groups and they disappeared within 24 h.

The most common adverse events were dizziness or headache, nausea or vomiting in group O and dizziness or headache, drowsiness, blurred or double vision in group I (Figure 2).

The adverse event durations were mostly D1 or D2 in groups I and O (57.1% vs. 40.1% and 38.6% vs. 38.2%), although there were more patients that adverse events were continued to D3 in group O (4.29 vs. 16.64), D3 was very small (Figure 3).

### 3.3. Adverse Events according to Sedative Type

In total, 153 patients received PK25, 71 received PK20, and 39 received propofol. Patients who received PK25 had a significantly higher incidence of adverse events at T1 than patients who received PK20 and propofol (34.6% vs. 21.1% vs. 17.9%, *p* = 0.012). The incidence of adverse events at T2 and T3 were 15.7% and 3.3% in patients who received PK25, 9.9% and 14.1% in patients who received PK20, and 10.3% and 2.6% in patients who received propofol; the difference among groups was not significant (Table 3).

### 3.4. The Correlation between Additional Bolus and Adverse Events Occurrence

The additional bolus administration of sedatives was shown to be significantly higher in propofol in both groups (Figure 4)

However, when we compared the adverse events with additional bolus and without additional bolus in each group, there were no relationships between additional bolus and adverse events (Figure 5).

## 4. Discussion

Procedural sedation for pediatric patients is applied in diverse fields using various methods [2]. Sedative agents are selected according to the invasiveness and duration of the procedure. The safety and efficacy of these sedatives have been reported [17,18,19], but there is still no standard protocol. The most commonly used sedative agents for pediatric patients are midazolam, ketamine, and propofol [18,20,21], although dexmedetomidine is increasingly being used [13,22]. Ketamine produces dissociative sedation, resulting in deeply depressed consciousness while generally maintaining airway reflexes [23]. It may also cause agitation, tremor, and excessive salivation, which disturbs respiration [24,25,26]. We use propofol or a propofol–ketamine combination for procedural sedation in pediatric patients. In the case of deep sedation, propofol is preferred because of its fast onset and short duration and because it causes less postoperative nausea and vomiting [27,28,29]. However, proper management is required because it may result in hemodynamic instability and respiratory depression [30,31]. The propofol–ketamine combination is more effective than propofol or ketamine alone [32,33,34]. 

Immediate post-procedural adverse effects under sedation include headache, dizziness, nausea, and vomiting [9,10,11,12,13]. In this study, the overall incidence rate of adverse events was 48.3% in group I and 33.6% in group O. In group I, most adverse events occurred within 2 h after the procedure; dizziness or headache and drowsiness were the most common symptoms, consistent with previous studies [9,10]. Nausea or vomiting and dizziness or headache was the most frequent adverse events during the early post-procedural period (between 2 h and 12 h). In group O, on the other hand, most of the adverse events occurred in the early post-procedural period after the procedure, and dizziness or headache and nausea or vomiting were the most common symptoms in that period. Averse events 12 h after the procedure were rare in group I and although rare, nausea or vomiting was occurred with high frequency in group O. In addition, about 90% of adverse events did not last more than 12 h. And the most frequent side effects after sedation; nausea, vomiting, and dizziness were not lasting for more than 12 h. 

In this study, we did not include respiratory depression and fall down in the checklist. In our protocol, patients were transferred to the sedation recovery room only after the anesthesiologist confirmed that the patient had no respiratory depression. Moreover, patients must lie in bed for 2 h to promote hemostasis in BME cases and for 4 h to prevent headache in ITC cases. Therefore, the patients are under close observation during that period. As a result, both respiratory depression and fall down had not occurred after procedural sedation in patients with hematologic malignancies in our previous experiences.

The incidence of adverse events was considered not to be related to the additional bolus administration. As can be seen from the results, the additional administration was high in propofol, but, the incidence of adverse events showed no increase according to the addition. Adverse events in the acute phase were least frequent in patients who received only propofol, and most frequent in those receiving PK25. Previous studies reported that a combination of ketamine and propofol is more effective than using either agent alone for procedural sedation [14,32,33]. However, the results of this study showed that T1 adverse effects were higher in the PK25 group which was more commonly used in inpatients. In other words, the higher the proportion of ketamine, the more side effects appeared, indicating that higher doses of ketamine may be more likely to trigger adverse events [26,34]. Most of those studies focused on the safety of the propofol-ketamine combination in terms of hemodynamic and respiratory side effects, as well as the recovery time from sedation [14,35,36]. Our study focused on the adverse events that may occur continuously after procedural sedation, not those that may occur during procedural sedation.

Interestingly, most adverse events appeared at an early stage in outpatients, as opposed to immediately after the procedure in inpatient. There are two possible reasons for this. First, adverse events in inpatients were recorded by a nurse in charge of their patients from recovery, whereas in outpatients, descriptions by their parents, thus, there was a difference in expertise with respect to those who observed the adverse events. Second, inpatients were transferred directly from the recovery room to the ward, while outpatients were discharged home. Adverse events may not have been properly observed even if they occurred on the way home, and the means of transportation after discharge or movement distance may also have influenced the variation of occurrence time and duration of adverse events. Meanwhile, as the adverse events did not last for more than 12 h in most cases and no intervention were necessary in both groups, these sedatives can be safely applied even in outpatients. 

### Study Limitations

A limitation of this study was its retrospective nature. Although we found differences in the onset time of adverse events between inpatients and outpatients, the patients were not randomized, and there were group differences in demographic characteristics, particularly age. In addition, the type of sedative agent administered was not randomized, which could have influenced the results. Nevertheless, this was the first study to evaluate the adverse events associated with procedural sedation up to 24 h after the procedure in pediatric patients who had undergone BME or ITC.

## 5. Conclusions

Most adverse events after BME and ITC, such as dizziness or headache, drowsiness, and nausea or vomiting occurred within 2 h after the procedure in inpatient children, compared to 2 or 12 h after the procedure in outpatient children. Most adverse events did not last longer than 12 h. A propofol–ketamine combination with a higher proportion of ketamine seems to produce more adverse events within 2 h after the procedure regardless of the additional administration of the sedatives. Overall, propofol and a propofol-ketamine combination seem to be safe even in pediatric outpatients undergoing BME and ITC. A prospective randomized study is needed to validate the safety of these sedative agents and determine the optimal proportion of ketamine when using it in combination with propofol in order to strengthen the reliability of this study.

## Figures and Tables

**Figure 1 medicina-58-01208-f001:**
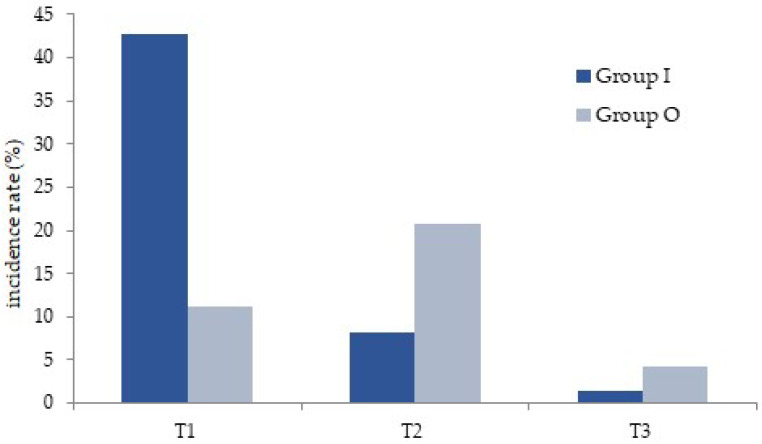
The incidence rates of adverse events. T1: acute period (<2 h); T2: early period (<12 h); T3: late period (<24 h).

**Figure 2 medicina-58-01208-f002:**
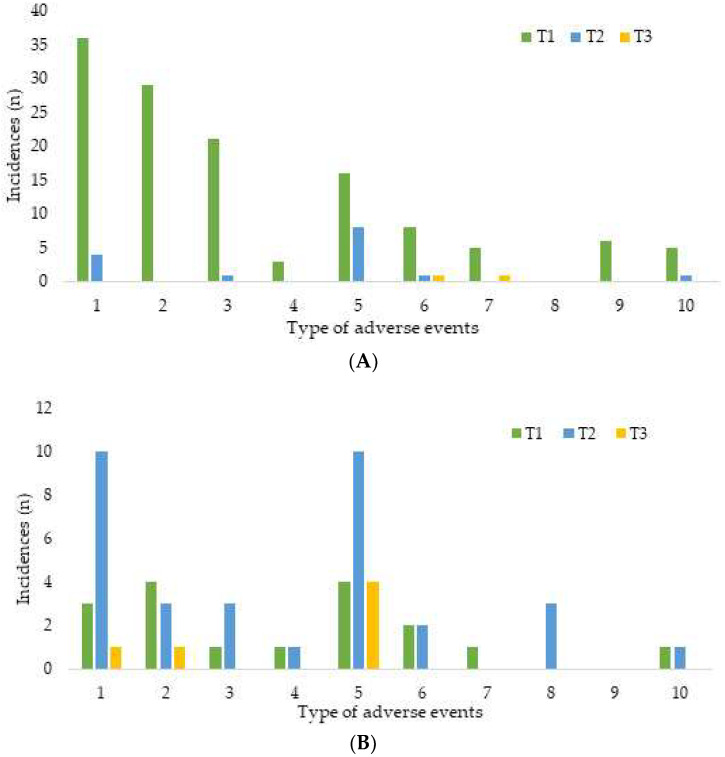
Distribution of adverse event types in (**A**) group I and (**B**) group O. 1: dizziness or headache; 2: drowsiness; 3: blurred or double vision; 4: unusual smell; 5: nausea or vomiting; 6: skin itching; 7: nystagmus or tremor; 8: insomnia or nightmare; 9: hallucination or confusion; 10: crying or irritability, anxiety.

**Figure 3 medicina-58-01208-f003:**
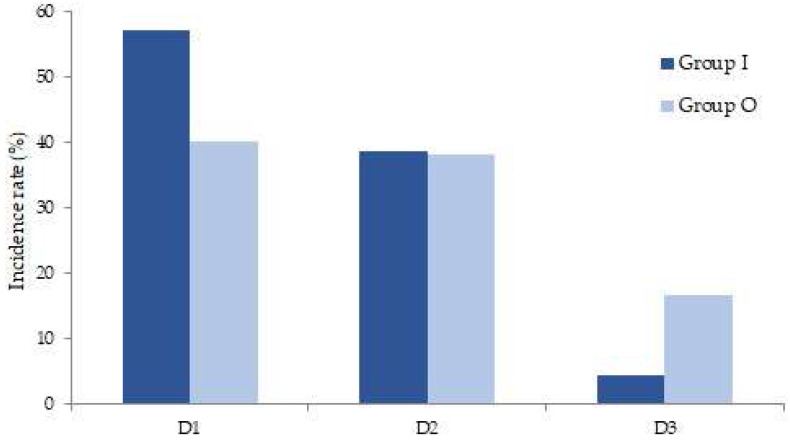
Duration of adverse events. D1, 2 h; D2, 2–12 h; D3, 24 h.

**Figure 4 medicina-58-01208-f004:**
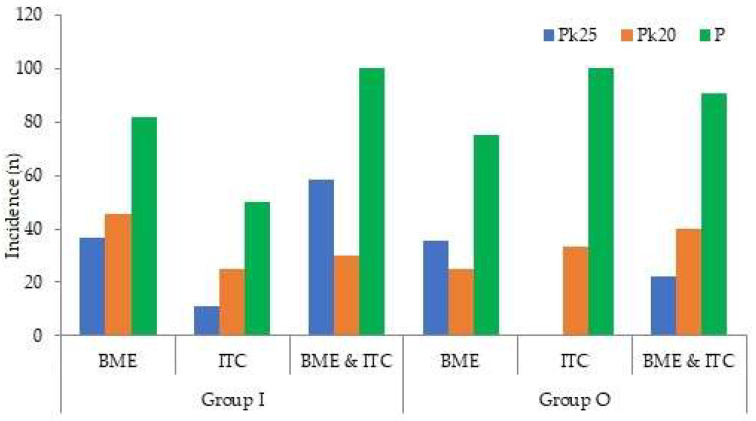
The proportion of additional bolus of each sedative in both groups.

**Figure 5 medicina-58-01208-f005:**
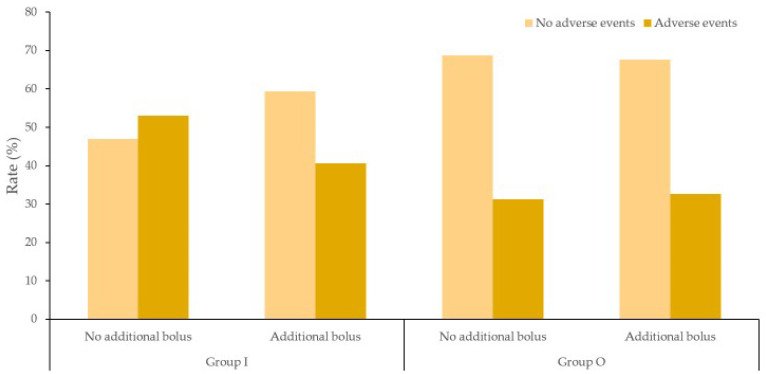
The comparison of adverse events occurrence with additional bolus and without additional bolus in each group.

**Table 1 medicina-58-01208-t001:** Adverse events checklist.

Adverse Event		Yes	No	Start Time	Finish Time	Timing *	Duration ^†^
1	Dizziness or headache						
2	Drowsiness						
3	Blurred or double vision						
4	Unusual smell						
5	Nausea and/or vomiting						
6	Skin itching						
7	Nystagmus or tremor						
8	Insomnia, nightmares						
9	Hallucination (confusion)						
10	Crying, irritability, anxiety						

* timing of event (acute = T1; early = T2; late = T3); ^†^ duration of event (acute = D1; early = D2; late = D3).

**Table 2 medicina-58-01208-t002:** Patient characteristics.

	Group I (n = 147)	Group O (n = 116)	*p* Value
Age (years)	9.1 ± 4.4	7.0 ± 4.3	<0.001
Sex (F) (n(%))	78 (53%)	60 (48%)	0.901
BMI	17.6 ± 3.4	17.7 ± 3.7	0.852
Duration of sedation (min)	15.6 ± 9.0	10.9 ± 5.4	<0.001
Duration of procedure (min)	13.3 ± 8.1	9.2 ± 5.4	<0.001
Type of sedation (n(%), PK25/PK20/P)	107 (72.8)/25 (17)/15 (10.2)	46 (39.7)/46 (39.7)/24 (20.6)	<0.001
Type of procedure (n(%), BME/ITC/combined)	74 (50.3)/24 (16.3)/48 (32.7)	56 (48.3)/11 (9.5)/49 (42.2)	0.141
Type of hematological malignancy (n(%), ALL/AML)	95 (64.6)/52 (35.4)	80 (68.9)/36 (31.1)	0.46
Fentanyl administered (µg/kg)	1.5 ± 1.2	1.7 ± 1.2	0.687

Categorical variables are shown as numbers, and other variables are shown as mean ± standard deviation. BMI: body mass index; Pk25: ratio of 3:1: PK20: ratio of 4:1; BME: bone marrow examination; ITC: intrathecal chemotherapy.

**Table 3 medicina-58-01208-t003:** Incidence rates of adverse events according to sedative type.

	PK25 (n = 153)	PK20 (n = 71)	P (n = 39)	*p* Value
T1 (n, [%])	53 (34.6)	15 (21.1)	7 (17.9)	0.012
T2 (n, [%])	24 (15.7)	7 (9.9)	4 (10.3)	0.487
T3 (n, [%])	5 (3.3)	1 (14.1)	1 (2.6)	0.051

PK25, propofol-ketamine combination at a ratio of 3:1 (propofol 7.5 mg and ketamine 2.5 mg in 1 mL); PK20, propofol-ketamine combination at a ratio of 4:1 (propofol 8 mg and ketamine 2 mg in 1 mL); *p*, propofol; T1, 2 h after the procedure (acute); T2, 12 h after the procedure (early); T3, 24 h after the procedure (delayed).

## Data Availability

The data presented in this study are available on request from the corresponding author. The data are not publicly available due to privacy due to privacy.

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
