# Peer review of "Delayed Adverse Events after Procedural Sedation in Pediatric Patients with Hematologic Malignancies"

_medicina, 2022, doi:10.3390/medicina58091208_

Round 1

Reviewer 1 Report (Previous Reviewer 1)

The authors have improved the manuscript and it will be of interest to readers.  The only additional suggestion would be to modify the following important sentence in the abstract for clarity:  "Patients who received propofol-ketamine combination treatment had a significantly higher incidence of adverse events at T1".  I would suggest "Patients with a higher proportion of ketamine in a propofol-ketamine mixture had a significantly higher proportion of adverse events at T1 (34.6%), as compared with those with a mixture with a lower proportion of ketamine (21.1%) or propofol alone (17.9%) (p=0.012). "

Author Response

Thank you very much for reviewing my article.

I answered the parts you pointed out has been corrected and marked in red through the article.

Point 1: The only additional suggestion would be to modify the following important sentence in the abstract for clarity:  "Patients who received propofol-ketamine combination treatment had a significantly higher incidence of adverse events at T1".  I would suggest "Patients with a higher proportion of ketamine in a propofol-ketamine mixture had a significantly higher proportion of adverse events at T1 (34.6%), as compared with those with a mixture with a lower proportion of ketamine (21.1%) or propofol alone (17.9%) (p=0.012). "

Response 1: The part mentioned in abstract has been modified.

Reviewer 2 Report (Previous Reviewer 2)

I have no further comments for the Authors.

Author Response

Thank you very much for your positive comment after careful review.
Your opinion will give my thesis an opportunity to bring meaningful results.

This manuscript is a resubmission of an earlier submission. The following is a list of the peer review reports and author responses from that submission.

Round 1

Reviewer 1 Report

 Thank you for the opportunity to review the manuscript “Delayed adverse events after procedural sedation in pediatric patients with hematologic tumors”.  This topic is of much interest to practitioners who provide procedural sedation for such children, relevant to selection of medications, duration of post-procedural monitoring, and instructions to families after post-procedural monitoring.  The study has a substantial number of patients and appropriately evaluates both time frame until adverse events as well as duration of the events.  The most significant issue with the manuscript is the events evaluated, which are incomplete both in scope and definition yet also so numerous that it is confusing to the reader what is most relevant.

More specific critiques are elaborated by section below.

1)     Title:  generally “tumors” is used to indicate “masses”, would suggest changing to “malignancies”

2)     Abstract: The phrase “In total, 20 adverse events were recorded” sounds like this is a measured number of events total in the study as opposed to the methods.  Would suggest “20 adverse events were assessed”

3)     Abstract:  Include “single center” and the time frame

4)     Introduction:   Would provide a reference if possible (even if not in the peds heme/onc population) regarding what sedation meds are used in children.  Would also delete the sentence “The most commonly used sedative agent in children is midazolam” since this may no longer be true, especially in this population.

5)     Methods:  Put the inclusion criteria first and make them more clear:  Children 2years-<15 years of age who underwent BME or ITC under sedation/anesthesia (without invasive airway?  In sedation suite?) over the time frame 2019-2022

6)     Methods:  postoperative management: for inpatients, you mean “discharged” from the recovery area to the inpatient floor?  Would spell this out.  For outpatients, what was the “time allotted for hemostasis and headache prevention” – same as for inpatients or how was this decided?  That seems really long if they were kept for 4 hours prior to discharge home. 

7)     Methods/results:  The adverse events are the biggest issue with the study.  Some critiques/questions related to this:

a)      How did you choose this list?  How did you choose the time frames?  Some references and rationale is needed.

b)     How were the adverse events recorded – prospectively or retrospectively?  Were there clear criteria or checklists, or was this from narrative notes?

c)      For adverse events, hypoxia needs to be included.  Even though it is less likely after the procedure, it is the most feared and universally reported adverse effect of procedural sedation.   Hemodynamic instability similarly is more likely during the procedure but could be present in the immediate post procedural period as well. 

d)     Generally the other adverse events included in such studies, especially with ketamine, are hallucinations and/or agitation, and nausea/vomiting.  With propofol I would also worry about excessive sleepiness causing injury such as falls.  The list you have provided is too long and does not clearly/cleanly have those categories, while having other elements that overlap and some that are not really relevant and not present at all.  My suggestions would be to pare down to 10 categories (or fewer!) and have a “hallucinations” category, and somehow indicate whether “excessive drowsiness” caused any falls or other real problems.  Perhaps you could combine nausea/vomiting, combine blurred/double vision, eliminate categories that are irrelevant and had no incidence such as fever and tremor, and create new categories of “hallucinations” (perhaps combining nightmares with other categories if appropriate).

8)     Results:  Table 2

a.      For sex, would just pick one gender to report and compare the number and percentage.  Such as Sex=female   Group I  78 (53%)    Group O 60 (48%)

b.      The height and weight can be removed, they are different because the age is different but the BMI shows the important info

c.      For fentanyl, mcg/kg would be more informative than total mcg

d.      For type of sedation and type of procedure, need percentages.  This may be relevant to why PK25 had increased T1 adverse events, since it appears that the I group was more likely to get that and they were likely sicker

9)     Results:  Figure 1 is unnecessary and can be removed.  You could consider making a T1/T2/T3 figure instead analogous to Fig 3 instead (but then Table 3 with this information could be removed)

10)  Results:  some more clinical information about how/why the I and O groups were different would be very helpful.  What types of malignancies did they have, where were they in their care, why were they inpatients, did they have any other organ failure?  This would provide context as to why the I group had more adverse events. 

11)  Results:  Similarly, it would be helpful to know whether any of the adverse events required intervention if that data are available.  For outpatients – did they have to stay in clinic after the procedure, or return to the ED or clinic?  For inpatients – did the care team have to intervene?

12)  Discussion:  precedex and midazolam are not included in the study so do not need to be in the discussion

13)  Discussion:  When discussing the PK25 having the most adverse effects, need to put in context that inpatients also received more PK25 (73% of inpatients got this vs 40% of outpatients if I’m doing the proportions correctly) and were likely systematically different from the outpatients (sicker, one assumes).  Why did the I group get PK25 more frequently?   Given this systematic difference, the conclusion that the adverse events were due to PK25 cannot be made.  

Author Response

Thank you very much for reviewing my article and providing detailed comments.

I answered your comments as follows; and the parts you pointed out has been corrected and marked in blue through the article.

We checked and corrected all the parts you mentioned. First of all, the checklist items have been organized to express more clearly and concisely, and the accompanying tables and figures have been modified according to the changed results. In order to convey the precise message of this article, parts other than those you pointed out are marked in orange

Title

Point1: generally “tumors” is used to indicate “masses”, would suggest changing to “malignancies”

Response 1: I changed it as you mentioned

Abstract

Point2:  The phrase “In total, 20 adverse events were recorded” sounds like this is a measured number of events total in the study as opposed to the methods.  Would suggest “20 adverse events were assessed”

Response 2: I modified this part as you mentioned

Point3: Include “single center” and the time frame

Response 3: I included a phrase ‘single center’ and duration of study period as you mentioned.

Introduction

Point4: Would provide a reference if possible (even if not in the peds heme/onc population) regarding what sedation meds are used in children.  Would also delete the sentence “The most commonly used sedative agent in children is midazolam” since this may no longer be true, especially in this population.

Response 4: The sentence “the most ~ is midazolam” was deleted and reference regarding what used sedative agent in children was inserted.

Methods

Point5: Put the inclusion criteria first and make them more clear:  Children 2years-<15 years of age who underwent BME or ITC under sedation/anesthesia (without invasive airway?  In sedation suite?) over the time frame 2019-2022

Response 5: As you mentioned, we modified the inclusion criteria by adding anesthetic method and hematological malignancy categories in detail at 2.1 study population section and table 1.

Point6: (1) postoperative management: for inpatients, you mean “discharged” from the recovery area to the inpatient floor?  Would spell this out.  

Response 6-(1): we modified postoperative management section as you mentioned.

(2) For outpatients, what was the “time allotted for hemostasis and headache prevention” – same as for inpatients or how was this decided?  That seems really long if they were kept for 4 hours prior to discharge home. 

Response 6-(2): While inpatients can continue to observe their condition in ward next to recovery room in supine position, outpatients are unavailable to maintain supine position on the way home. In other words, they need to consider the time to complete hemostasis in bone marrow examination site (for 2 hours) and intrathecal chemo spread time and prevention headache (for 4 hours).==> mentioned in method section, postoperative management part of 2.2 procedure and intervention. Therefore, they lied down at least 2 hours to maximal 4 hours, and they stayed in hospital for 4 hour for observation.

Methods/results:  

Point7: The adverse events are the biggest issue with the study.  Some critiques/questions related to this:

  1. a) How did you choose this list? How did you choose the time frames?  Some references and rationale is needed.

 Response 7-a: It was written based on the known side effects of sedatives as well as the adverse reactions (such as unusual smell, tremor, confusion and anxiety) that patients complained for the previously in our hospital.

There were no physical injuries such as falls.

The time frame was classified into according to the following criteria;

an acute phase (T1): average time spent in recovery room considering the direct side effects of sedatives

an early phase (T2): period considering half-life of sedatives and residual drug elimination time of sedatives

a delayed phase (T3): period considering the duration of existing abnormal reaction in T1 or T2.

There was no previous study on using this time frame, so considering the duration of action of sedatives and after the occurrence of adverse reactions, a checklist (Table 1) was made and continuously used.

  1. b) How were the adverse events recorded – prospectively or retrospectively?  Were there clear criteria or checklists, or was this from narrative notes?

Response 7-b: In the electrical recovery record, we have separate checklists of the adverse events after deep sedation. From review of these check lists, table 1 check list was created retrospectively. We mentioned in 2.3 Data collection and evaluation part.

  1. c) For adverse events, hypoxia needs to be included.  Even though it is less likely after the procedure, it is the most feared and universally reported adverse effect of procedural sedation.   Hemodynamic instability similarly is more likely during the procedure but could be present in the immediate post procedural period as well. 

Response 7-c: The list of adverse events investigated was not intended to identify common adverse reactions of sedatives such as hemodynamic instability, respiratory failure or hypoxia through apnea.

As you mentioned, hypoxia is a feared and dangerous side effect. However, during procedural sedation, an anesthesiologist provided breathing assistance with a disposable breathing circuit (so called, Jackson Ree breathing circuit : Mapleson F system) and continuously supplies 3~5L/min oxygen through a facial mask until this patient was ready to leave the recovery room, so, there was no case of hypoxia due to respiratory depression.

Therefore, hypoxia, which is included in general side effects, was not included in the investigated adverse events. In particular, hypoxia was an item that couldn’t be classified as a side effect of sedatives in the process of confirming delayed adverse reactions.

In all case, after spontaneous respiration is confirmed, they were transferred to recovery room. If hypoxia or hemodynamic instability had occurred, that patient would not have been transferred recovery room and not have been evaluated this checklist.

  1. d) Generally the other adverse events included in such studies, especially with ketamine, are hallucinations and/or agitation, and nausea/vomiting.  With propofol I would also worry about excessive sleepiness causing injury such as falls.The list you have provided is too long and does not clearly/cleanly have those categories, while having other elements that overlap and some that are not really relevant and not present at all.  My suggestions would be to pare down to 10 categories (or fewer!) and have a “hallucinations” category, and somehow indicate whether “excessive drowsiness” caused any falls or other real problems.  Perhaps you could combine nausea/vomiting, combine blurred/double vision, eliminate categories that are irrelevant and had no incidence such as fever and tremor, and create new categories of “hallucinations” (perhaps combining nightmares with other categories if appropriate).

Response 7-d: These pediatric patients, the subjects of this study, should lie down

without movement for at least 2 hours after the procedure. Handrail on the bed

should be raised and put a fall protection pad so that they did not move and fall.

In the case of tremor, it was sometimes observed together with nystagmus, so it was

included in these categories and combined with nystagmus.

Excessive drowsiness is when the child continues to complain that he is sleepy.

Under any circumstances, ‘fall’ was not happened and won’t be included in these categories. 

Adverse event

No

Adverse event

1

Dizziness

1,9

1

Dizziness or headache

2

drowsiness

2

2

drowsiness

3

blurred

3,4

3

blurred or double vision

4

double vision

5

4

unusual smell

5

unusual smell

6,7

5

Nausea or/ and vomiting

6

Nausea

10

6

skin itching

7

vomiting

8,11

7

Nystagmus or tremor

8

Tremor

13, 14

8

Insomnia, nightmares

9

headache

16

9

Hallucination (confusion)

10

skin itching

15,17,18

10

Crying or irritability, anxiety

11

Nystagmus

12

no or decreased appetite

12: no or decreased appetite-deleted

19: fever- deleted

20: others- deleted

13

insomnia

14

nightmares

15

depression, sadness, irritability

16

Confusion->halluciation

17

anxiety

18

crying

19

Fever

20

other

As your advice, we modified to the table on the right above.

 Tremor occurred in pediatric patients after sedation by ketamine, so it was presented in

  1. 7 nystagmus as a side effect of Ketamine.

Results

Point 8: table 2

  1. For sex, would just pick one gender to report and compare the number and

percentage.  Such as Sex=female   Group I  78 (53%)    Group O 60 (48%)

Response 8-a: we changed this as you mentioned         

  1. The height and weight can be removed, they are different because the age is different

but the BMI shows the important info

Response 8-b: we changed this as you mentioned

  1. For fentanyl, mcg/kg would be more informative than total mcg

Response 8-c: We modified it as you mentioned.

  1. For type of sedation and type of procedure, need percentages.  This may be relevant to why PK25 had increased T1 adverse events, since it appears that the I group was more likely to get that and they were likely sicker

Response 8-d: We added % in the type of sedation and the type of procedure in Table 2.

Point 9: Figure 1 is unnecessary and can be removed. You could consider making a T1/T2/T3 figure instead analogous to Fig 3 instead (but then Table 3 with this information could be removed)

Response 9: we removed figure 1 as you mentioned. I put in the Figure 1 as T1/T2/T3 as

you mentioned.

Point 10: some more clinical information about how/why the I and O groups were

different would be very helpful.  

What types of malignancies did they have, 

Response 10: hematologic malignancies are various types. We classified into the following two types by focusing on main pediatric hematologic malignancy; â‘ ALL: Acute lymphoblastic leukemiaa and â‘¡ AML: Acute myelocytic leukemia. These two types are presented in detail in the context.

In both groups, two types of hematological malignancies were included. There was no difference in disease types between the two groups.

where were they in their care, why were they inpatients

Response 10: There is no special standard chosen in group I.

At first, whether to proceed with BME or ICT is absolutely decided by the pediatrician

(with hemato-oncologic specialist). And then the choice of admission (selected in

group I) and outpatient (group O) is the patients and their parents’ preference. In

other words, inclusion in group I has nothing to do with whether or not

chemotherapy is undergone. Being an inpatient doesn’t mean being admitted to do

something else like chemotherapy or other procedures.

The medical condition does not affect the choice of inpatient or outpatient.

did they have any other organ failure?  

This would provide context as to why the I group had more adverse events. 

Response 10: Pediatric patients with organ failure are not included in this assessment. They were excluded in this evaluation to avoid confounding factors and described in method section, study population part.

Point 11: a Similarly, it would be helpful to know whether any of the adverse events required intervention if that data are available.  

Response 11-a: None of the above adverse events require special intervention. There were no management even if adverse events were occurred and we describe this in result section, 3.2 adverse events in each group.

  1. For outpatients – did they have to stay in clinic after the procedure, or return to the ED or clinic?  For inpatients – did the care team have to intervene?

Response 11-b: all patients need to stay without movement.

The outpatients have to stay in sedation recovery room for hemostasis or intrathecal

chemo absorption, and then discharged to their home. While in case of inpatients, the

main observer of adverse events were their parents because of discharge, in case of

inpatient, they transferred to their ward and main adverse events’ observer was nurse.

Neither group did any intervene for adverse events regardless of medical staff or not.

Discussion

Point 12: precedex and midazolam are not included in the study so do not need to be in the discussion

Response 12: We deleted the sentences related to Midazolam explanation as you

mentioned. Dexmedetomidine is left because it is included in the description of

currently used pediatric sedatives.

Point 13: When discussing the PK25 having the most adverse effects, need to put in context that inpatients also received more PK25 (73% of inpatients got this vs 40% of outpatients if I’m doing the proportions correctly) and were likely systematically different from the outpatients (sicker, one assumes).  Why did the I group get PK25 more frequently?   Given this systematic difference, the conclusion that the adverse events were due to PK25 cannot be made

Response 13: There is no difference between inpatient and outpatient sedative types. There was a difference in the timing of adverse events, and the sedative that caused them high was PK25. As suggested in the study limitation, although it was not intended, the type of sedative is not the same, so a prospective study through the present study report is needed.

Reviewer 2 Report

The topic proposed by the authors is of great interest. Knowledge about which adverse events (AEs) after procedural sedation, following the discharge from the recovery room up to 24 h, may occur, of what severity, and how often are still lacking. Well-conducted and in-depth studies are undoubtedly helpful and necessary.

In my opinion, however, the method chosen by the authors should have been more rigorous to obtain valid data and, therefore, reliable results.

See below, in my opinion, some major critical issues:

The choice of the study population

The pediatric patient with hematologic tumors often is submitted to procedural sedation. However, the pathology, ongoing therapies, and clinical conditions concerning the stage of the disease can alone cause most of the adverse events recorded in this study. This element cannot be underestimated as a confounding factor.

Type of sedative agents administered

Administration of fentanyl initial bolus, and additional boluses when necessary, should be evaluated as a further risk factor in the appearance of adverse events, especially at T1.

How many patients required additional boluses, and whether there was a difference between those who received only the initial bolus and those who also received additional ones should be specified in the AEs recorded. Why do the authors administer two types of propofol-ketamine combination (3:1, 4:1)? What determined the choice of one ratio over the other? Whatever the answer is, it should be specified. Furthermore, in the introduction and discussion, reference is made to midazolam which is not among the drugs used in this study. I suggest removing all considerations to midazolam.

Data collection group O

It is written that for patients in group O researcher recorded the adverse events observed by the caregiver. The study is retrospective and covers a time period of approximately 3 years (March 2019-February 2022). It is imperative to detail how the information was collected from caregivers, otherwise, the reliability of the results is difficult to support/demonstrate. (how did the caregivers remember these "minor" events after some time passes?). This is especially emphasized considering that in the results the group O incidence rates of Aes at T2 and at T3 have been respectively 24.1% (the highest percentage for the group) and 4.3%.

Statistical analysis

I am not a statistician, but I wonder if the variables under study could not have been the subject of multivariate analysis. Otherwise, the choice of analysis performed by the authors should be better specified.

Author Response

Thank you very much for reviewing my article and providing detailed comments.

I answered your comments as follows; and the parts you pointed out has been corrected and marked in green through the article.

We checked and corrected all the parts you mentioned. First of all, the checklist items have been organized to express more clearly and concisely, and the accompanying tables and figures have been modified according to the changed results. In order to convey the precise message of this article, parts other than those you pointed out are marked in orange

The choice of the study population

Point1: The pediatric patient with hematologic tumors often is submitted to procedural sedation. However, the pathology, ongoing therapies, and clinical conditions concerning the stage of the disease can alone cause most of the adverse events recorded in this study. This element cannot be underestimated as a confounding factor.

Response 1: The side effects may occur due to the condition of the underlying disease in patients with hematologic malignancy, but, in order to prevent possible interference with confounding factor, patients who developed side effects or abnormal condition before deep sedation were excluded and described in method section. In addition, among the 20 adverse events, no. 12 ; no or decreased appetite and no. 19: fever are excluded because it can be vague to conclude as adverse events caused by sedation itself. the Table 1, the adverse events checklist is modified based on the side effects that may occur with the sedatives.

Type of sedative agents administered

Point2: Administration of fentanyl initial bolus, and additional boluses when necessary, should be evaluated as a further risk factor in the appearance of adverse events, especially at T1. How many patients required additional boluses, and whether there was a difference between those who received only the initial bolus and those who also received additional ones should be specified in the AEs recorded.

Response 2: in order to find out the relationship between the administration of the additional bolus and the occurrence of adverse events, 3.4 The correlation between additional bolus and adverse events occurrence was added to the result section.

Point3: Why do the authors administer two types of propofol-ketamine combination (3:1, 4:1)? What determined the choice of one ratio over the other?

Whatever the answer is, it should be specified.

Response 3: We inserted the description of selecting three types of sedatives in Method section, 2.2 procedure ad intervention, anesthetic method part.

Point4: Furthermore, in the introduction and discussion, reference is made to midazolam which is not among the drugs used in this study. I suggest removing all considerations to midazolam.

Response 4: We deleted the sentences related to midazolam in this article as you mentioned.

Data collection group O

Point5: It is written that for patients in group O researcher recorded the adverse events observed by the caregiver. The study is retrospective and covers a time period of approximately 3 years (March 2019-February 2022). It is imperative to detail how the information was collected from caregivers, otherwise, the reliability of the results is difficult to support/demonstrate. (how did the caregivers remember these "minor" events after some time passes?). This is especially emphasized considering that in the results the group O incidence rates of Aes at T2 and at T3 have been respectively 24.1% (the highest percentage for the group) and 4.3%.

Response 5: Here, the caregivers in Group O are their parents and added in method section 2.3 in detail. In particular, due to the characteristics of patients’ diseases, intensive management is required, so their mother closely monitors and manages their children. Although the reported datum were records for three years, those which were collected from the caregiver, were the results right at that time after discharge confirmed by phone two times. Since we collected and managed all records in this way, they were not inaccurate results after a long time by the memory of their parents. Therefore, we were able to check the detailed answers of checklist and the occurrence time of it.

Statistical analysis

Point6: I am not a statistician, but I wonder if the variables under study could not have been the subject of multivariate analysis. Otherwise, the choice of analysis performed by the authors should be better specified.

Response 6: In this study, we did not use multivariate analysis concerning confounding factors. Instead, we presented in the 2.4 statistical analysis section in detail. Since this observational study is not a comparison through the same randomization between the two groups, a study that considers this should be conducted in the future.

Round 2

Reviewer 1 Report

The authors have improved the manuscript significantly, however additional minor issues remain as follows:

1) Methods:  Still unclear how the primary outcome was collected.  For inpatients, it is stated that recovery nurses recorded adverse events - was this true even 24 hours later when they were on the ward?  For outpatients, did the researcher call the parent on the phone, and if so how long after discharge?

2) Since both the type and duration of sedation were significantly different in inpatients (Table 2), and T1 adverse effects were higher in the PK25 group (Table 4) which was more commonly used in inpatients, it would appear that the patients who got more ketamine for longer had the most adverse effects and this is the primary explanation for the results.  This should be moved earlier in the discussion to make this more clear.  Also the phrase "The incidence of adverse events was considered not to be related to the additional bolus administration or type of sedatives" in the discussion does not appear accurate since type of sedatives (ie PK25 vs propofol) was related to incidence of adverse effects in Table 4.

3) The authors' practice of extended 2-4 hour recovery time is not universal.  It should be pointed out in the discussion that this likely contributed to the lack of any observed respiratory depression after discharge from recovery.  Also I still recommend mentioning that no falls were reported, since the bedrails wouldn't be protecting the outpatients after discharge. 

Author Response

Response of Reviewer 1 comments

Thank you very much for reviewing my article and giving good comments.

I answered your comments as follows; and the parts you pointed out has been corrected and marked in blue through the article.

Point 1:

Methods:  Still unclear how the primary outcome was collected.  For inpatients, it is stated that recovery nurses recorded adverse events - was this true even 24 hours later when they were on the ward?  For outpatients, did the researcher call the parent on the phone, and if so how long after discharge?

Response 1: The observation process of adverse events after recovery of group I and O was modified and described in text in method section. Inpatients (Group I) were transferred to their room in the ward when fully awake and were allowed to lie in bed for 2 h to promote hemostasis in BME cases and for 4 h to prevent headache in ITC cases. Outpatients (Group O) were discharged and went to their homes after the period for hemostasis and headache prevention, identical to the patients in Group I. Adverse events were recorded by the ward nurse in charge of the patient in Group I, while the researcher recorded the adverse events of the patients in Group I described by the caregiver, i.e., their parents, on the phone until 24 h after discharge. (line on 98~102, 109~112)

Point 2-1:

Since both the type and duration of sedation were significantly different in inpatients (Table 2), and T1 adverse effects were higher in the PK25 group (Table 4) which was more commonly used in inpatients, it would appear that the patients who got more ketamine for longer had the most adverse effects and this is the primary explanation for the results.  This should be moved earlier in the discussion to make this more clear. 

Response 2-1: We described in the earlier part of discussion section as you mentioned. (line on 246~ 250)

Point 2-2:

Also the phrase "The incidence of adverse events was considered not to be related to the additional bolus administration or type of sedatives" in the discussion does not appear accurate since type of sedatives (ie PK25 vs propofol) was related to incidence of adverse effects in Table 4.

Response 2-2: We deleted ‘type of sedatives’ in order to make the content clear.

Point 3:  

The authors' practice of extended 2-4 hour recovery time is not universal.  It should be pointed out in the discussion that this likely contributed to the lack of any observed respiratory depression after discharge from recovery.  Also I still recommend mentioning that no falls were reported, since the bedrails wouldn't be protecting the outpatients after discharge. 

Response 3: The duration of recovery is presented by referring to your comments on the postoperative management of the method section. Recovery time of up to 4 hours is solely due to the nature of the procedure and it is not associated with the occurrence of respiratory complications. Recovery begins after the anesthesiologists confirm that respiration is fully recovered (line on 96~98).

In addition, the item ‘no fall’ was also described in discussion section. However, in both groups, after an intensive recovery period of up to 4 hours, their movements are unrestricted and free, and also they do not have to stay in bed. Therefore, falls have never occurred as an adverse event of sedatives (line on 232~239).

Reviewer 2 Report

I think the authors have addressed and revised their manuscript according to my comments and suggestions.

In my opinion, the manuscript has been sufficiently improved to warrant publication in Medicina.

Author Response

Thank you very much for your careful review of my article and for your positive comments. Your comments have given opportunities to make my article more accurate and meaningful.